# Asymmetric biomimetic transamination of α-keto amides to peptides

Weiqi Cai[1,2], Xuelong Qiao[1,2], Hao Zhang[1], Bo Li [1], Jianhua Guo[1], Liangliang Zhang[1], Wen-Wen Chen[1] & Baoguo Zhao [1✉]

Peptides are important compounds with broad applications in many areas. Asymmetric transamination of α-keto amides can provide an efficient strategy to synthesize peptides, however, the process has not been well developed yet and still remains a great challenge in both enzymatic and catalytic chemistry. For biological transamination, the high activity is attributed to manifold structural and electronic factors of transaminases. Based on the concept of multiple imitation of transaminases, here we report N-quaternized axially chiral pyridoxamines **1** for enantioselective transamination of α-keto amides, to produce various peptides in good yields with excellent enantio- and diastereoselectivities. The reaction is especially attractive for the synthesis of peptides made of unnatural amino acids since it doesn't need great efforts to make chiral unnatural amino acids before amide bond formation.

[1] The Education Ministry Key Lab of Resource Chemistry and Shanghai Key Laboratory of Rare Earth Functional Materials, Shanghai Normal University, Shanghai, P. R. China. [2]These authors contributed equally: Weiqi Cai, Xuelong Qiao. ✉email: zhaobg2006@shnu.edu.cn

P eptides are one type of the most important compounds with high biological activities, which are widely present in many natural products, pharmaceutically relevant molecules, and biological systems[1–3]. Especially in recent years, there appears a growing interest in therapeutic peptides[1–3] and more and more peptide drugs have been developed (Fig. 1a)[4–6]. Development of

alternative new methods for the synthesis of peptides is always highly desirable and potentially useful[7,8].

Enzymatic transamination is an important process to produce chiral amines such as amino acids in biological systems (Fig. 1b)[9,10], which is promoted by transaminases (Fig. 1c) with pyridoxal/pyridoxamine 5′-phosphates as the coenzyme[9–13]. Mimicking the biological process[14], i.e., asymmetric biomimetic transamination, affords

**Fig. 1 Peptides and transamination. a** Peptide drugs. **b** Biological transamination. **c** Catalyst design. **d** 1,3-Proton shift. **e** Transamination of α-keto amides.

**Fig. 2 Synthesis chiral pyridoxamines 1.** The detailed synthetic procedures have been presented in Supplementary Information (SI). THF = tetrahydrofuran, DCM = dichloromethane.

a highly intriguing method to synthesize $NH_2$-free amines from readily available carbonyl compounds[15–17]. The chemistry has attracted much attention since the 1970s[18–36]. The studies mainly include stoichiometric chiral pyridoxamine-promoted asymmetric transamination of α-keto acids[18–21], pyridoxal/pyridoxamine-catalyzed asymmetric transamination of α-keto acids[22–26], and chiral base/Lewis acid-catalyzed asymmetric transamination of α-keto esters and activated ketones[27–36]. Asymmetric transamination of α-keto amides can potentially provide an appealing new strategy to produce peptides. However, to the best of our knowledge, asymmetric transamination of α-keto amides to peptides are barely reported[37,38], although the reverse process, transamination of peptides at the N-termini to α-keto amides, has been widely developed and already have been successfully applied to protein modification[38–45]. In contrast to the transamination of α-keto acids, asymmetric transamination of α-keto amides to peptides remains a challenge for enzymatic catalysis likely due to non-naturally occurring process and it is also a challenge for chemical catalysis probably because the complicated structure of α-keto amides requires more active catalysts to promote transamination.

Previous studies have suggested that asymmetric 1,3-proton shift between the ketimine intermediate and the aldimine is likely a key step for biological transamination (Fig. 1d)[11–13,46,47]. In order to accelerate this step, evolution has elegantly optimized transaminases by incorporating a Lys residue at an appropriate position[48–50]. The ε-$NH_2$ group of the Lys residue serves as an intramolecular base to deprotonate the benzylic C-H of the ketimine (Fig. 1d)[48–50]. In addition, the p$K_a$ values of the pyridine N in the coenzyme pyridoxal 5′-phosphate (PLP) and pyridoxamine 5′-phosphate (PMP) are around 8.5[51–53], thus the pyridine N is predominantly protonated in biological systems (near pH 7)[53–56]. The strong electron-withdrawing property of the protonated pyridine ring helps to increase the acidity of the benzylic C-H bond (Fig. 1d)[54–58]. The two effects work together to promote the transformation from the ketimine to the aldimine via 1,3-proton shift, magically accelerating transamination process. Inspired by the controlled protonation of the pyridine N of PLP in biological systems, Rapoport has developed N-methylpyridinium-4-carboxaldehyde benzenesulfonate (Rapoport's salt) as an effective transamination reagent for conversion of amines to carbonyl compounds[59]. Francis have found that Rapoport's salt displays significantly improved efficiency in the transamination of proteins as compared to pyridinium-4-carboxaldehyde, converting the N-termini into the corresponding carbonyl groups[44,45]. Recently, we have proved that introducing an amine side arm to a chiral pyridoxamine can remarkably increase its activity and enantioselectivity for asymmetric transamination of α-keto acids[26] and also have observed that quaternization of the pyridine N of a chiral pyridoxal leads to dramatical improvement of catalytic activity for asymmetric biomimetic Mannich reaction[60] and aldol reaction[61] of glycinate. On the basis of the structural characteristics of transaminases[46–50,53–56] as well as the previously reported studies[26,44,45,59–62], we has designed N-quaternized biaryl axially chiral pyridoxamines 1 bearing an amine side arm, mimicking transaminases in multiple aspects for catalytic asymmetric transamination of α-keto amides to peptides (Fig. 1c). The quaternization always keeps the pyridine ring with strong electron-withdrawing ability to improve the benzylic C-H acidity of

the ketimine intermediate during transamination, no matter under acidic or basic conditions[60,61]. Like the Lys residue does in biological transamination, the amine side arm can serve as an intramolecular base to facilitate 1,3-proton shift.

Here we show that asymmetric biomimetic transamination of α-keto amides 2 can be achieved by using chiral pyridoxamines 1 as the catalyst, to produce various peptides 3 with excellent enantiopurities (Fig. 1e).

## Results

**Catalyst synthesis.** The synthesis of chiral pyridoxamines 1 started with reductive amination of compound 5[26] to introduce the amine side chain. Protecting the two amine groups with di-*tert*-butyl dicarbonate gave intermediates 6 (Fig. 2). Treatment of 6 with methyl iodide and subsequent deprotection with hydrochloric acid afforded N-methyl pyridoxamines 1a-e in good yields.

**Condition optimization.** With diphenylglycine (4) as the amine source[25,26,63], catalyst chiral pyridoxamine 1b was first tested for the transamination of glycinyl α-keto amide 2a (Fig. 3, entry 1). The originally-formed $NH_2$-free transamination product was treated with di-*tert*-butyl dicarbonate to avoid the cyclization to piperazinedione during the isolation[64], to give the corresponding N-Boc-protected dipeptide 3a in 20% yield with 76% ee. Additives have significant impacts on the reaction in terms of enantioselectivity and activity. Increased yield and enantioselectivity were obtained for transaminations performed in MeOH/$H_2O$ or TFEA/$H_2O$ with HOAc/KOAc or HOAc/$Na_2HPO_4$ as the additives (Fig. 3, entries 2 and 10 vs 3–9). Chiral pyridoxamine 1b exhibited the best performance among the catalysts 1a-e examined (Fig. 3, entries 10–14).

**Substrate scope.** Under the optimal conditions, various glycinyl α-keto amides containing alkyl (for 3b-e), aromatic (for 3a and 3f-i), or heteroatomic alkyl (for 3j-k) groups were all smoothly transaminated to give the corresponding N-Boc-protected glycinyl dipeptides 3a-k in 70–94% yields with up to 98% ee (Fig. 4). Chiral glycinyl α-keto amide (for 3 l) displayed excellent diastereoselectivity (98:2 dr). Transamination of α-keto phenylbutanamides of chiral amino acid esters produced various N-Boc-protected dipeptides 3m-y in 56–93% yields with up to 99:1 diastereoselectivity. Peptidyl α-keto amides were also effective for the asymmetric transamination, to form tripeptides 3z-ab and tetrapeptides 3ac-ad in 60–87% yields with excellent diastereoselectivities under very mild conditions. Various functional groups such as C-C double bond (for 3c, 3 l and 3 y), $NH_2$-sensitive bromide (for 3j), silyl group (for 3k), OH group of Tyr (for 3 s), NH group of Trp (for 3t), amide $CONH_2$ of Asn (for 3 v), Boc-protected Lys residue (for 3w), Boc-protected guanidine (for 3x), and basic $NH_2$ group of Lys (for 3ab) were all well tolerated by the transamination likely due to the mild reaction conditions.

In order to investigate the impacts of catalyst and substrates on diastereomeric induction, several representative α-keto amides (for 3m-n, 3r, 3v-y, and 3aa-ab) were examined respectively using (S)-

**Fig. 3 Investigation of reaction parameters.** TFEA = 2,2,2-trifluoroethanol. [a]Reaction conditions: **2a** (0.10 mmol), **4** (0.11 mmol), **1** (0.0050 mmol), HOAc (0.40 mmol), base (0.20 mmol) in solvent (0.48 mL) and $H_2O$ (0.12 mL) at 20 °C for 48 h unless otherwise stated. The reaction mixtures were then treated with di-*tert*-butyl dicarbonate (0.30 mmol) at rt for 3 h. [b]Isolated yields based on α-keto amide **2a**. [c]The ee values were determined by HPLC analysis. [d]Reaction time was 72 h.

| Entry | Reaction conditions | Yield (%)[b] | ee (%)[c] |
|---|---|---|---|
| 1 | (S)-**1b**, MeOH/H$_2$O | 20 | 76 |
| 2 | (S)-**1b**, HOAc/KOAc, MeOH/H$_2$O | 88 | 93 |
| 3 | (S)-**1b**, HOAc/KOAc, THF/H$_2$O | 21 | 84 |
| 4 | (S)-**1b**, HOAc/KOAc, CH$_3$CN/H$_2$O | 49 | 67 |
| 5 | (S)-**1b**, HOAc/KOAc, DCM/H$_2$O | 29 | 80 |
| 6 | (S)-**1b**, HOAc/KOAc, TFEA/H$_2$O | 77 | 95 |
| 7 | (S)-**1b**, HOAc/NaHCO$_3$, MeOH/H$_2$O | 29 | 89 |
| 8 | (S)-**1b**, HOAc/Na$_2$HPO$_4$, MeOH/H$_2$O | 88 | 95 |
| 9 | (S)-**1b**, HOAc/Et$_3$N, MeOH/H$_2$O | 66 | 89 |
| 10[d] | (S)-**1b**, HOAc/Na$_2$HPO$_4$, TFEA/H$_2$O | 88 | 98 |
| 11[d] | (S)-**1a**, HOAc/Na$_2$HPO$_4$, TFEA/H$_2$O | 77 | 77 |
| 12[d] | (S)-**1c**, HOAc/Na$_2$HPO$_4$, TFEA/H$_2$O | 86 | 92 |
| 13[d] | (R)-**1d**, HOAc/Na$_2$HPO$_4$, TFEA/H$_2$O | 57 | -84 |
| 14[d] | (R)-**1e**, HOAc/Na$_2$HPO$_4$, TFEA/H$_2$O | 80 | -85 |

**1b** (5 mol%), (*R*)-**1b** (5 mol%), and achiral pyridoxamine **7** (20 mol %) as the catalyst. The corresponding peptides were formed with *S* configurations of the newly generated chiral centers from catalyst (*R*)-**1b** and *R* configurations from (*S*)-**1b**. The chiral pyridoxamine catalyst dominated the stereoselectivity, while the chiral groups on the amino acid residues of α-keto amides throwed little influence on the diastereomeric induction probably due to being far away from the reaction centers as well as the flexibility of the skeletons of the α-keto amides. No matter which configuration of the catalyst **1b** was applied, excellent diastereoselectivities were always obtained, even for α-keto amides (for **3n** and **3aa**) with a nearby bulky chiral amino acid residue and for those that displayed obvious substrate-induction on diastereoselectivity in **7**-catalyzed non-asymmetric transamination (5:95 dr for **3r** and 19:81 dr for **3x**). For α-keto amide **2y** bearing two nearby chiral centers, a pair of diastereomers (*R*,*R*,*S*)-**3y** and (*R*,*S*,*S*)-**3y** were respectively obtained in good yields

with high enantiopurities by using (*S*)-**1b** and (*R*)-**1b** as the catalyst. The absolute configurations of the newly generated chiral centers of peptides **3** were assigned by analog, based on the X-ray analysis of **3d**, **3m**, and **3r** (also see Supplementary Figs. 1–3 in SI).

**Synthetic applications.** Divergent extending an additional amino acid unit from a central peptide is of great interest for peptide drug screening and bioactivity studies. The synthesis would be difficult when the extended unit is a commercially unavailable unnatural amino acid. The transamination process provides an efficient strategy for the amino acid extending. For example, starting from the benzyl ester of protease inhibitor Ubenimex (**8**)[65], condensation with α-keto acids and subsequent asymmetric transamination afforded a variety of enantiopure peptides **3ae-ai** with one more amino acid residue extended (Fig. 5a). The

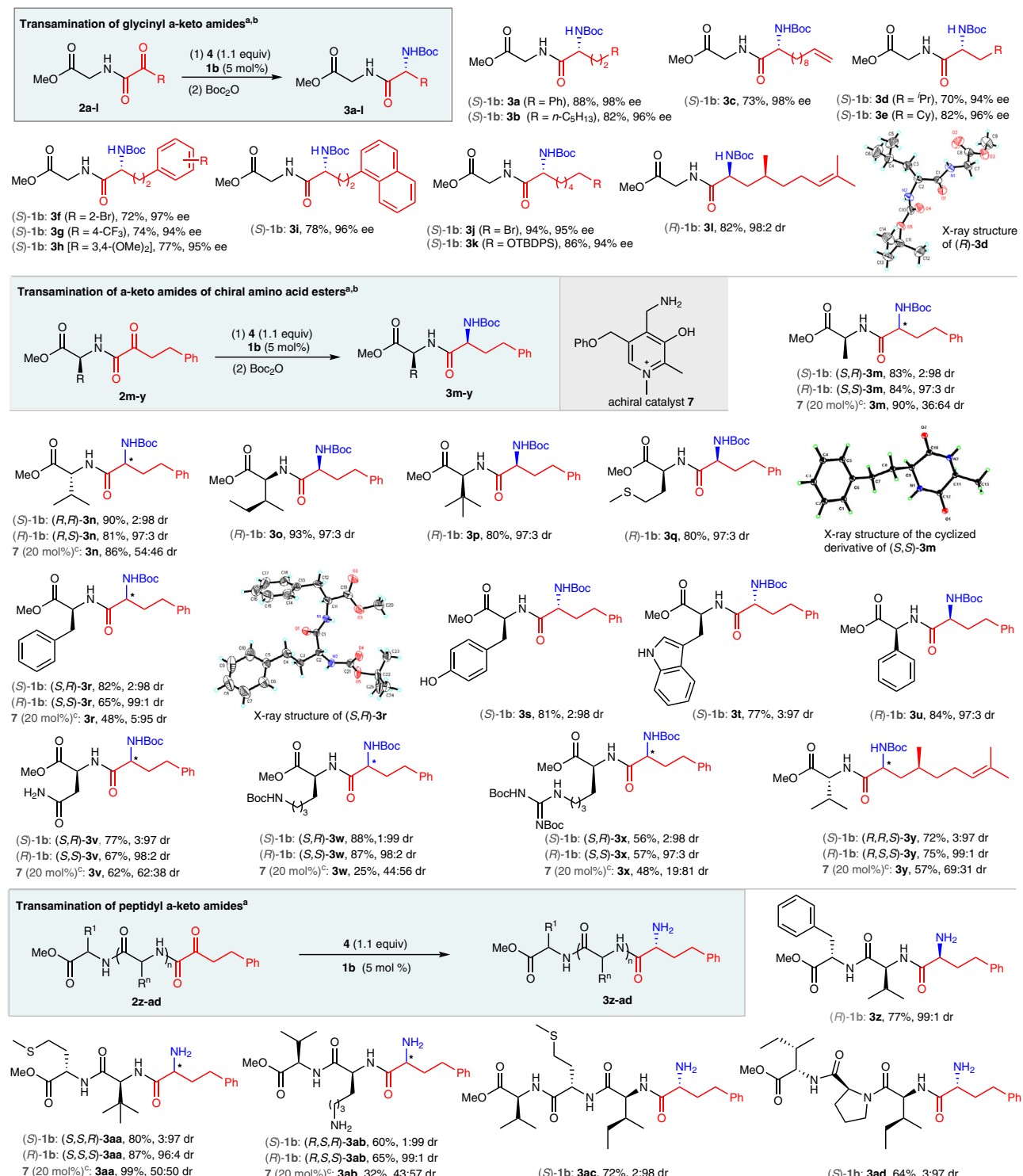

**Fig. 4 Asymmetric biomimetic transamination of α-keto amides.** TBDPS = *tert*-butyldiphenylsilyl. [a]Reaction conditions: **2** (0.10 mmol), **4** (0.11 mmol), **1b** (0.0050 mmol), HOAc (0.40 mmol), $Na_2HPO_4$ or KOAc (0.20 mmol) in $CF_3CH_2OH$ or MeOH (0.48 mL) and $H_2O$ (0.12 mL) for 48 or 72 h unless otherwise stated (See SI). For **3m-y**, the reactions were carried out in a double scale. Isolated yields were based on α-keto amides **2**. The ee and dr values were determined by HPLC analysis. [b]The crude reaction mixture was treated with di-*tert*-butyl dicarbonate (0.30 mmol) at room temperature for 3 h. [c]The reaction was carried out in a double scale with **7** (0.040 mmol, 20 mol %) at 50 °C for 72 h.

unprotected OH group remained untouched during the condensation and transamination.

Based on the "condensation-transamination" process, a new strategy for the synthesis of peptides also can be developed. As illustrated in Fig. 5b, the methyl ester of DPP-IV inhibitor Diprotin A (**10**)[66] underwent condensation with α-keto acid **9a**

and subsequent asymmetric transamination, forming tetrapeptide **3ad** with excellent diastereoselectivity. Repeating the reaction sequence two more times afforded hexapeptide **3ak** with high enantiopurity. The chirality of the extended amino acid residues was established along with the transamination process. The protocol is especially attractive for the synthesis of peptides made

**Fig. 5 Synthetic applications. a** Divergent synthesis of peptides. NMM = *N*-methylmorpholine. **b** Synthesis of peptides via successive transamination. EDCI = 1-(3-dimethylaminopropyl)-3-ethylcarbodiimide hydrochloride. HOBt = 1-hydroxybenzotriazole.

of unnatural amino acids, since it doesn't need great efforts to make NH₂-protected chiral unnatural amino acids before the amide bond formation.

**Reaction mechanism**. A plausible mechanism was proposed for the transamination (Fig. 6a)[11–13,25,26]. Pyridoxamine **1b** condenses with α-keto amide **2** to form ketimine **11**, which undergoes asymmetric 1,3-proton shift to aldimine **13** under the assistance of the amine side arm[46–50]. Hydrolysis of aldimine **13** releases peptide **3** and generates the pydridoxal, which is in situ converted into iminium **14** via intramolecular condensation. The iminium **14** then undergoes decarboxylative transamination with the amine source diphenylglycine (**4**) back to pyridoxamine catalyst **1b**[25,26,63], completing a catalytic cycle.

As expected, N-quaternization of the pyridine ring of the chiral pyridoxamines resulted in higher catalytic activity and better enantioselectivity for the asymmetric transamination (Fig. 6b,

**1b** vs **1g**). The stronger electron-withdrawing property makes the benzylic C-H of ketimine **11** more acidic and also stabilizes the corresponding delocalized carbanion **12** better[54–58,67–69], thus favoring the 1,3-proton shift and accelerating the transamination process. The control experiment confirmed the amazing effect of the amine side arm again (Fig. 6b, **1b** vs **1f**). Introducing an acetyl group onto the nitrogen to eliminate the basicity of the amine on the side arm led to marked decreases in activity and enantioselectivity. The amine side arm not only promotes the 1,3-proton shift by acting as an intramolecular base to deprotonate the benzylic C-H of ketimine **11** (Fig. 6a) but also helps to orient the α-keto amide by hydrogen bonding with the carbonyl oxygen of the amide group (Fig. 6c), resulting in improved activity and stereoselectivity[48–50]. Protonation of the delocalized carbanion **12** occurs at α−position of the amide group from the up side of the pyridine ring away from the amine side arm[26,46,47], to form the newly generated chiral center with *S* configuration from catalyst (*R*)-**1b**.

**Fig. 6 Reaction mechanism. a** Proposed reaction pathway. **b** Comparison of catalysts. **c** Proposed transition state for the asymmetric 1,3-proton shift.

## Discussion

In summary, based on the concept of multiple imitation of transaminases, we have developed N-quaternized axially chiral pyridoxamines **1** containing an amine side arm. With pyridoxamine **1b** as the catalyst, challenging substrates α-keto amides

were successfully transaminated to peptides in good yields with excellent enantio- and diastereoselectivities. The catalyst dominated the diastereoselective control for the transamination of chiral α-keto amides. Thus, a pair of diastereomeric peptides could be respectively obtained with high enantiopurities by

switching the configuration of the pyridoxamine catalyst. The strong electron-withdrawing property of the N-quaternized pyridine ring together with the cooperative catalysis of the amine side arm account for the increased catalytic activity and selectivity of the pyridoxamine **1b** in the transamination. The reaction can provide an efficient strategy for divergent and successive extension of peptides via condensation-transamination reaction sequence, which is especially attractive for the synthesis of peptides made of unnatural amino acids.

## Methods

**General procedure for the asymmetric biomimetic transamination Reaction (Fig. 4).** A mixture of α-keto amide **2** (0.10 mmol), chiral pyridoxamine **1b** (0.0050 mmol), 2,2-diphenylglycine (**4**) (0.11 mmol), HOAc (0.40 mmol), $Na_2HPO_4$ or KOAc (0.20 mmol), $CF_3CH_2OH$ or MeOH (0.48 mL), and $H_2O$ (0.12 mL) was stirred at 16–25 °C for the specified time. For glycinyl α-keto amides (for **3a-l**) and α-keto phenylbutanamides of amino acid esters (for **3m-y**), the crude reaction mixtures were treated with di-*tert*-butyl dicarbonate (0.3 mmol) at room temperature for 3 h after the transamination was completed, then concentrated via rotary evaporator to remove most of the solvent and isolated by column chromatography on silica gel with a mixed solvent ethyl acetate and petroleum ether as the eluant to give the products N-Boc-protected dipeptides **3a-y**. For Peptidyl α-keto amides (for **3z-ad**), the reaction mixtures were submitted to concentration via rotary evaporator to remove most of the solvent and then isolated by column chromatography on silica gel with a mixed solvent of dichloromethane, methanol and ammonia solution in ethanol (2.9 M) as the eluant to give the transamination products tripeptides **3z-aa** and tetrapeptides **3ab-ad** without $NH_2$-protection. The ee and dr values of **3a-ac** were determined by HPLC analysis.

## Data availability

The authors declare that the data supporting the findings of this study are available within the article and Supplementary Information file, or from the corresponding author upon reasonable request. For the experimental procedures, characterization data, and NMR spectra along with HPLC chromatograms, see Supplementary Information. The X-ray crystallographic coordinates for structures reported in this study have been deposited at the Cambridge Crystallographic Data Centre (CCDC), under deposition numbers of CCDC 2036531 (**3d**), CCDC 2036532 (the cyclized derivative of **3 m**), and CCDC 2036529 (**3r**). These data can be obtained free of charge from The Cambridge Crystallographic Data Centre via https://www.ccdc. cam.ac.uk/structures/.

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

## Acknowledgements

We are grateful for the generous financial support from the National Natural Science Foundation of China (21672148, 21871181), the Shanghai Municipal Education Commission (2019-01-07-00-02-E00029), the Shanghai Municipal Committee of Science and Technology (18ZR1447600, 20JC1416800), "111" Innovation and Talent Recruitment Base on Photochemical and Energy Materials (D18020), and Shanghai Engineering Research Center of Green Energy Chemical Engineering (18DZ2254200).

## Author contributions

B.Z. conceived and directed the project and wrote the paper. W.C and X.Q. conducted most of the experiments including the synthesis of the chiral pyridoxamine catalysts and the development of the asymmetric biomimetic transamination reaction. H.Z. synthesized some pyridoxamine intermediates and several α-keto amides for the transamination. B.L. performed some experiments for the catalyst development. J.G. and L.Z synthesized several α-keto amides for the transamination. W.C. revised the manuscript and the Supplementary Information.

## Competing interests

The authors declare no competing interests.
