## [Peer Review File · Nature Communications]

Reviewers' Comments:

Reviewer #1:

Remarks to the Author:

In this manuscript, Cai et al describe new chiral catalysts that are capable of promoting asymmetric transamination reactions, in a manner similar to that which is used by biological transaminase enzymes. In general, this paper does not represent a significant advance from prior work by the same group as well as other groups who have published on methylated pyridines that can perform similar chemistry. Thus, I do not find this work suitable for publication in a broad journal like Nature Communications, though I think, with minor revision, it should be considered at a more specialized journal.

- 1) One thing that strikes me from the very beginning is the use of the term "chiral peptides". With the exception of poly-Gly, I really can't imagine a peptide that isn't chiral, so I am confused by what the authors are trying to get at with this term. Does it mean a different type of chirality than is found naturally (D vs L), or is it just redundant?
- 2) The past work in the area of asymmetric biomimetic transamination is not well-described. The authors state that "asymmetric transamination of α -keto amides can potentially provide an appealing new strategy to produce chiral peptides, but the process has not been well developed yet and it still remains a formidable challenge in both biological and chemical catalytic systems". One of the references cited is from 1943, which I would not characterize as recent, and the other is a single report from 2014. It would be helpful to clarify what the true advances are in this paper, by accurately describing the recent work (of which I am certain there is some) in this area. For instance – why is it a "formidable challenge" and why has it "not been well developed yet"?
- 3) There is an error in Fig. 1d – the aldimine has an R group instead of what I believe should be an H.
- 4) The authors state that enzymes catalyze transamination by "protonating the pyridine nitrogen of the coenzyme pyridoxal/pyridoxamine". This is a minor point but should probably be re-phrased – the pKa of the pyridine N in PLP is around 8.5, meaning that it is predominantly protonated in biological systems near pH 7, so it seems more likely to be due to the chemical structure & resulting properties.
- 5) The authors have demonstrated that placing an amine side arm nearby helps to promote asymmetric transamination in a prior publication (JACS 2016). Thus, in this work, it seems the major advance is methylation of the pyridine N, which makes the catalyst more active. This piece has already been extensively published on by Matt Francis's group (<https://pubs.acs.org/doi/abs/10.1021/ja509955n>, <https://www.ncbi.nlm.nih.gov/pmc/articles/PMC4136391/>), and has also been reported much earlier by Rapoport (<https://pubs.acs.org/doi/pdf/10.1021/ja00380a019>) though there are no citations about either of these contributions. That past work focused on transamination that converts protein N-termini to 1,2 dicarbonyls, but it is the same reaction in reverse. Given this, I question if the chemical advance in this work is high impact enough to be suitable for this journal.
- 6) The authors demonstrate that these catalysts can be used to extend peptides. In general, I agree that there might be some applications where it would be potentially more tractable to do a two step process like the one described, particularly for unnatural amino acids that might not be commercially available. That being said, how well does this methodology tolerate nearby free side chain functional groups – the examples shown are largely aliphatic (except for one hydroxyl group) and wouldn't be expected to compete or participate in the reaction. It would be critical to see for side chains from Lys, Arg, Cys, Asp etc...
- 7) If the axial chirality is reversed, is the other stereoisomer for transamination obtained? I would have loved to see that as a demonstration for how versatile this approach is. If I am reading the structures correctly, it seems that the chirality that is obtained is more like what you would get for a D-amino acid rather than L?
- 8) There are a number of grammatical and language issues throughout the abstract and manuscript ("e.g. "abroad applications"). These should be fixed prior to publication

Reviewer #2:

Remarks to the Author:

Previously (JACS, 2016, 138, 10730) the authors (as well as other groups) introduced highly selective organocatalytic methods to carry out enantioselective transaminations of α -ketoesters. This is arguably one of the most versatile approaches to convert simple prochiral precursors in valuable α -amino acid derivatives.

In continuation of their own impressive contributions, the authors herein succeeded in developing a protocol to carry out the transamination of complex α -ketoamides with a carefully optimized catalyst structure. This method turned out to be highly selective and, very impressively, the authors succeeded in carrying out transaminations of highly functionalized small peptide precursors giving access to (non-natural) small peptides in a unique fashion. The whole paper is very illustratively written and the methods reported herein will for sure be of high interest to the community. Thus, in general I recommend publication of this article in a top journal.

One thing that I however missed when reading the paper were details on the influence of the catalyst configuration in the transamination of the diastereotopic starting materials (di- or tri-peptide precursors). Is it possible to access both diastereomeric pairs by switching the catalyst configuration or is there a pronounced match/mismatch scenario depending on starting materials and catalyst configuration?! This should be discussed in more detail and I also recommend discussing the diastereoselectivity (substratecontrol) of reactions of these chiral precursors when using achiral catalysts.

Reviewer #3:

Remarks to the Author:

The manuscript "Asymmetric Biomimetic Transamination of α -Keto Amides to Chiral Peptides" demonstrated the application of axially chiral pyridoxamines catalyzed biomimetic transamination in chiral peptide synthesis. The design of the axially chiral pyridoxamines bearing an amine side arm was inspired from enzyme structure, and yet it's quite meaningful to again function like enzymes to artificially synthesize diversified peptides, especially the potentially useful peptides with unnatural amino acids. In the biomimetic transamination catalyzed by axially chiral pyridoxamine catalysts, the chiral peptides were obtained with good yield and high enantioselectivity/diastereoselectivity, with good tolerance to functional groups in substrates.

In Fig. 3, for 3a-3k, the configuration of the chiral center bearing -NHBoc is (R). However, for 3l with a chiral methyl group at the γ -position, the configuration on the newly generated chiral center bearing -NHBoc is found to be opposite (i.e, S). Similar observation was found in the second part of Fig. 3, the configurations of the chiral centers bearing -NHBoc for 3m-3n and 3r-3t are (R); but for 3o-3q and 3u, they are (S). It looks that the side chain on the neighboring amino acid has a great impact on the selectivity of the biomimetic transamination. The authors are advised to discuss and explain this observation probably by elaborating the proposed mechanism in Fig. 5a.

For chiral amino acid esters 2m-u bearing a chiral center, diastereoselectivity is observed using chiral catalyst (S)-1b in the reaction. Considering the possible matched/mismatched transition states in the chiral induction step, would the use of chiral catalyst with opposite chirality (i.e. (R)) affect the diastereoselectivity of the reaction?

Did the authors study substrates bearing chiral amino acid unit together with substrate like 3l with a chiral methyl group at the γ -position? As both the chiral amino acid and chiral methyl group at the γ -position would affect the diastereoselectivity, it would be interesting to investigate the combined effects of them. Higher, lower or similar diastereoselectivity could be obtained?

In Fig. 3, the authors have investigated different structures of α -keto amides and different amino acids with diversified side chains, and several peptides have been studied, presenting good tolerance to C-C double bond, NH₂-sensitive bromide, silyl group, OH and NH groups. As presented in Fig. 3, 3q with a methionine unit can be resulted with good yield and high diastereoselectivity (80% yield, 97:3 dr). Is this reaction compatible with other amino acid side chains, for example, would a cysteine unit with a free thiol group be tolerated?

REVIEWER COMMENTS

Reviewer #1 (Remarks to the Author):

In this manuscript, Cai et al describe new chiral catalysts that are capable of promoting asymmetric transamination reactions, in a manner similar to that which is used by biological transaminase enzymes. In general, this paper does not represent a significant advance from prior work by the same group as well as other groups who have published on methylated pyridines that can perform similar chemistry. Thus, I do not find this work suitable for publication in a broad journal like Nature Communications, though I think, with minor revision, it should be considered at a more specialized journal.

1) One thing that strikes me from the very beginning is the use of the term “chiral peptides”. With the exception of poly-Gly, I really can’t imagine a peptide that isn’t chiral, so I am confused by what the authors are trying to get at with this term. Does it mean a different type of chirality than is found naturally (D vs L), or is it just redundant?

2) The past work in the area of asymmetric biomimetic transamination is not well-described. The authors state that “asymmetric transamination of α -keto amides can potentially provide an appealing new strategy to produce chiral peptides, but the process has not been well developed yet and it still remains a formidable challenge in both biological and chemical catalytic systems”. One of the references cited is from 1943, which I would not characterize as recent, and the other is a single report from 2014. It would be helpful to clarify what the true advances are in this paper, by accurately describing the recent work (of which I am certain there is some) in this area. For instance – why is it a “formidable challenge” and why has it “not been well developed yet”

3) There is an error in Fig. 1d – the aldimine has an R group instead of what I believe should be an H.

4) The authors state that enzymes catalyze transamination by “protonating the pyridine nitrogen of the coenzyme pyridoxal/pyridoxamine”. This is a minor point but should probably be re-phrased – the pKa of the pyridine N in PLP is around 8.5, meaning that it is predominantly protonated in biological systems near pH 7, so it seems more likely to be due to the chemical structure & resulting properties.

5) The authors have demonstrated that placing an amine side arm nearby helps to promote asymmetric transamination in a prior publication (JACS 2016). Thus, in this

work, it seems the major advance is methylation of the pyridine N, which makes the catalyst more active. This piece has already been extensively published on by Matt Francis's group (<https://pubs.acs.org/doi/abs/10.1021/ja509955n>, <https://www.ncbi.nlm.nih.gov/pmc/articles/PMC4136391/>), and has also been reported much earlier by Rapoport (<https://pubs.acs.org/doi/pdf/10.1021/ja00380a019>) though there are no citations about either of these contributions. That past work focused on transamination that converts protein N-termini to 1,2 dicarbonyls, but it is the same reaction in reverse. Given this, I question if the chemical advance in this work is high impact enough to be suitable for this journal.

6) The authors demonstrate that these catalysts can be used to extend peptides. In general, I agree that there might be some applications where it would be potentially more tractable to do a two step process like the one described, particularly for unnatural amino acids that might not be commercially available. That being said, how well does this methodology tolerate nearby free side chain functional groups – the examples shown are largely aliphatic (except for one hydroxyl group) and wouldn't be expected to compete or participate in the reaction. It would be critical to see for side chains from Lys, Arg, Cys, Asp etc...

7) If the axial chirality is reversed, is the other stereoisomer for transamination obtained? I would have loved to see that as a demonstration for how versatile this approach is. If I am reading the structures correctly, it seems that the chirality that is obtained is more like what you would get for a D-amino acid rather than L?

8) There are a number of grammatical and language issues throughout the abstract and manuscript ("e.g. "abroad applications"). These should be fixed prior to publication

Reviewer #2 (Remarks to the Author):

Previously (JACS, 2016, 138, 10730) the authors (as well as other groups) introduced highly selective organocatalytic methods to carry out enantioselective transaminations of α -ketoesters. This is arguably one of the most versatile approaches to convert simple prochiral precursors in valuable α -amino acid derivatives.

In continuation of their own impressive contributions, the authors herein succeeded in developing a protocol to carry out the transamination of complex α -ketoamides with a carefully optimized catalyst structure. This method turned out to be highly selective and, very impressively, the authors succeeded in carrying out transaminations of highly

functionalized small peptide precursors giving access to (non-natural) small peptides in a unique fashion. The whole paper is very illustratively written and the methods reported herein will for sure be of high interest to the community. Thus, in general I recommend publication of this article in a top journal.

One thing that I however missed when reading the paper were details on the influence of the catalyst configuration in the transamination of the diastereotopic starting materials (di- or tri-peptide precursors). Is it possible to access both diastereomeric pairs by switching the catalyst configuration or is there a pronounced match/mismatch scenario depending on starting materials and catalyst configuration?! This should be discussed in more detail and I also recommend discussing the diastereoselectivity (substratecontrol) of reactions of these chiral precursors when using achiral catalysts.

Reviewer #3 (Remarks to the Author):

The manuscript “Asymmetric Biomimetic Transamination of α -Keto Amides to Chiral Peptides” demonstrated the application of axially chiral pyridoxamines catalyzed biomimetic transamination in chiral peptide synthesis. The design of the axially chiral pyridoxamines bearing an amine side arm was inspired from enzyme structure, and yet it’s quite meaningful to again function like enzymes to artificially synthesize diversified peptides, especially the potentially useful peptides with unnatural amino acids. In the biomimetic transamination catalyzed by axially chiral pyridoxamine catalysts, the chiral peptides were obtained with good yield and high enantioselectivity/diastereoselectivity, with good tolerance to functional groups in substrates.

In Fig. 3, for 3a-3k, the configuration of the chiral center bearing -NHBoc is (R). However, for 3l with a chiral methyl group at the γ -position, the configuration on the newly generated chiral center bearing -NHBoc is found to be opposite (i.e, S). Similar observation was found in the second part of Fig. 3, the configurations of the chiral centers bearing -NHBoc for 3m-3n and 3r-3t are (R); but for 3o-3q and 3u, they are (S). It looks that the side chain on the neighboring amino acid has a great impact on the selectivity of the biomimetic transamination. The authors are advised to discuss and explain this observation probably by elaborating the proposed mechanism in Fig. 5a.

For chiral amino acid esters 2m-u bearing a chiral center, diastereoselectivity is

observed using chiral catalyst (S)-1b in the reaction. Considering the possible matched/mismatched transition states in the chiral induction step, would the use of chiral catalyst with opposite chirality (i.e. (R)) affect the diastereoselectivity of the reaction?

Did the authors study substrates bearing chiral amino acid unit together with substrate like 3l with a chiral methyl group at the γ -position? As both the chiral amino acid and chiral methyl group at the γ -position would affect the diastereoselectivity, it would be interesting to investigate the combined effects of them. Higher, lower or similar diastereoselectivity could be obtained?

In Fig. 3, the authors have investigated different structures of α -keto amides and different amino acids with diversified side chains, and several peptides have been studied, presenting good tolerance to C-C double bond, NH₂-sensitive bromide, silyl group, OH and NH groups. As presented in Fig. 3, 3q with a methionine unit can be resulted with good yield and high diastereoselectivity (80% yield, 97:3 dr). Is this reaction compatible with other amino acid side chains, for example, would a cysteine unit with a free thiol group be tolerated?

POINT-TO-POINT RESPONSE

Response to Reviewer #1: Great thanks for the helpful comments and suggestions

(1) *One thing that strikes me from the very beginning is the use of the term “chiral peptides”. With the exception of poly-Gly, I really can’t imagine a peptide that isn’t chiral, so I am confused by what the authors are trying to get at with this term. Does it mean a different type of chirality than is found naturally (D vs L), or is it just redundant?*

Response: We have changed all the terms “chiral peptide” appeared in the main text and SI to “peptide”.

(2) *The past work in the area of asymmetric biomimetic transamination is not well-described. The authors state that “asymmetric transamination of α -keto amides can potentially provide an appealing new strategy to produce chiral peptides, but the process has not been well developed yet and it still remains a formidable challenge in both biological and chemical catalytic systems”. One of the references cited is from 1943, which I would not characterize as recent, and the other is a single report from 2014. It would be helpful to clarify what the true advances are in this paper, by accurately describing the recent work (of which I am certain there is some) in this area. For instance – why is it a “formidable challenge” and why has it “not been well developed yet”*

Response: We have added the discussion on the reverse process, i.e. the transamination from peptides at the N-termini to α -keto amides. Although the transamination of peptides to α -keto amides has been widely developed and already successfully applied to protein modification, to the best of our knowledge, transamination of α -keto amides to peptides are barely reported. The examples that we found include transamination of pyruvylalanine with α -aminophenylacetic acid as the amine source. (*J. Biolog. Chem.* **1943**, *147*, 541-547) and transamination of N-terminal α -ketoamides with stoichiometric pyridoxamine as the amine source (*Chem. Res. Toxicol.* **2014**, *27*, 637-648). Asymmetric transamination of α -keto amides to peptides is a challenge for enzymatic catalysis likely due to non-naturally occurring process and it is also a challenge for chemical catalysis probably because the complicated structure of α -keto amides requires more active catalysts to promote transamination. The corresponding discussion has been added to the main text and the leading references on transamination of peptides to α -keto amides have been cited.

(3) *There is an error in Fig. 1d – the aldimine has an R group instead of what I believe should be an H.*

Response: According to Reviewer 1's suggestion, we have corrected the error in Figure 1d. The R group has been changed to H. The figure has been updated.

(4) *The authors state that enzymes catalyze transamination by “protonating the pyridine nitrogen of the coenzyme pyridoxal/pyridoxamine”. This is a minor point but should probably be re-phrased – the pKa of the pyridine N in PLP is around 8.5, meaning that it is predominantly protonated in biological systems near pH 7, so it seems more likely to be due to the chemical structure & resulting properties.*

Response: We have changed the description on the protonated status of the pyridine nitrogen of the coenzyme pyridoxamine/pyridoxal. As suggested, the previous description “protonating the pyridine nitrogen of the coenzyme pyridoxal/pyridoxamine” has been changed to “the pK_a values of the pyridine N in the coenzyme pyridoxal 5'-phosphate (PLP) and pyridoxamine 5'-phosphate (PMP) are around 8.5, thus the pyridine N is predominantly protonated in biological systems (near pH 7)”. The related references have been cited.

(5) *The authors have demonstrated that placing an amine side arm nearby helps to promote asymmetric transamination in a prior publication (JACS 2016). Thus, in this work, it seems the major advance is methylation of the pyridine N, which makes the catalyst more active. This piece has already been extensively published on by Matt Francis's group (<https://pubs.acs.org/doi/abs/10.1021/ja509955n>, <https://www.ncbi.nlm.nih.gov/pmc/articles/PMC4136391/>), and has also been reported much earlier by Rapoport (<https://pubs.acs.org/doi/pdf/10.1021/ja00380a019>) though there are no citations about either of these contributions. That past work focused on transamination that converts protein N-termini to 1,2 dicarbonyls, but it is the same reaction in reverse. Given this, I question if the chemical advance in this work is high impact enough to be suitable for this journal.*

Response: We have added the discussion on the transamination from amines to carbonyl compounds by using *N*-methylpyridinium-4-carboxaldehyde as the amine acceptor. In 1982, Rapoport developed *N*-methylpyridinium-4-carboxaldehyde benzenesulfonate (Rapoport's salt) as an effective transamination reagent for deamination of amines to carbonyl compounds (*J. Am. Chem. Soc.* **104**, 4446-4450). In

recent years, Francis found that Rapoport's salt displayed significantly improved efficiency in the transamination of proteins as compared to pyridinium-4-carboxaldehyde, converting the N-termini into the corresponding carbonyl groups (*J. Am. Chem. Soc.* **135**, 17223-17229; *J. Am. Chem. Soc.* **137**, 1123-1129). The corresponding references have been cited.

(6) *The authors demonstrate that these catalysts can be used to extend peptides. In general, I agree that there might be some applications where it would be potentially more tractable to do a two step process like the one described, particularly for unnatural amino acids that might not be commercially available. That being said, how well does this methodology tolerate nearby free side chain functional groups – the examples shown are largely aliphatic (except for one hydroxyl group) and wouldn't be expected to compete or participate in the reaction. It would be critical to see for side chains from Lys, Arg, Cys, Asp etc...*

Response: We have investigated more substrates bearing different functional side chains. The transamination not only well tolerated with functional groups such as NH₂-sensitive bromide (for **3j**), OH group of Tyr (for **3s**) and NH group of Trp (for **3t**), but also displayed good activity and diastereoselectivity with α -keto amides containing a nearby Boc-protected Lys residue (for **3w**), Boc-protected guanidine (for **3x**), or basic NH₂ group of Lys (for **3ab**). The information and the corresponding discussion have been added to the main text. Figure 3 in the main text has been updated. The substrate synthetic procedures, characterization data of these α -keto amides **2** and their transamination products **3** along with NMR spectra have been included in SI.

α -Keto amide **3al** containing a nearby Cys side chain was not obtained due to

intramolecular cyclization to form compound methyl (*R*)-5-oxo-6-(2-phenylethylidene)thiomorpholine-3-carboxylate (**3al-cycled**).

(7) *If the axial chirality is reversed, is the other stereoisomer for transamination obtained? I would have loved to see that as a demonstration for how versatile this approach is. If I am reading the structures correctly, it seems that the chirality that is obtained is more like what you would get for a D-amino acid rather than L?*

Response: Several representative α -keto amides were transaminated respectively using (*S*)-**1b** and (*R*)-**1b** as the catalyst. The corresponding peptides were formed with *S* configurations from catalyst (*R*)-**1b** and *R* configurations from (*S*)-**1b**. The chiral pyridoxamine catalyst dominated the stereoselectivity, while the chiral groups on the amino acid residues threw little influence on the diastereomeric induction probably due to being far away from the reaction center as well as the flexibility of the skeletons of the α -keto amides. No matter which configuration of the catalyst was applied, excellent diastereoselectivities were always obtained, even for α -keto amides (for **3n** and **3aa**) with a nearby bulky chiral amino acid residue. The information and the corresponding discussion have been added to the main text. Figure 3 in the main text has been updated. The corresponding information also have been included in SI.

(8) *There are a number of grammatical and language issues throughout the abstract and manuscript (“e.g. “abroad applications”). These should be fixed prior to publication.*

Response: We have checked the manuscript carefully and corrected typo errors including “abroad applications” revised to “broad applications”, “decarbonate” changed to “dicarbonate” and etc.

Response to Reviewer #2: Great thanks for the helpful comments and suggestions

(1) *One thing that I however missed when reading the paper were details on the influence of the catalyst configuration in the transamination of the diastereotopic starting materials (di- or tri-peptide precursors). Is it possible to access both diastereomeric pairs by switching the catalyst configuration or is there a pronounced match/mismatch scenario depending on starting materials and catalyst configuration?! This should be discussed in more detail and I also recommend discussing the diastereoselectivity (substratecontrol) of reactions of these chiral precursors when using achiral catalysts.*

Response: we have investigated the impacts of catalyst and substrates on diastereomeric induction. Several representative α -keto amides were transaminated respectively using (*S*)-**1b** (5 mol%), (*R*)-**1b** (5 mol%), and achiral pyridoxamine **7** (20 mol%) as the catalyst. The corresponding peptides were formed with *S* configurations from catalyst (*R*)-**1b** and *R* configurations from (*S*)-**1b**. The chiral pyridoxamine catalyst dominated the stereoselectivity, while the chiral groups on the amino acid residues threw little influence on the diastereomeric induction probably due to being far away from the reaction center as well as the flexibility of the skeletons of the α -keto amides. No matter which configuration of the catalyst was applied, excellent diastereoselectivities were always obtained, even for α -keto amides (for **3n** and **3aa**) with a nearby bulky chiral amino acid residue and for those that displayed obvious substrate-induction on diastereoselectivity in **7**-catalyzed non-asymmetric transamination (5:95 dr for **3r** and 19:81 dr for **3x**). The information and the corresponding discussion have been added to the main text. Figure 3 in the main text has been updated. The corresponding information also have been included in SI.

Response to Reviewer #3: Great thanks for the helpful comments and suggestions

(1) *In Fig. 3, for 3a-3k, the configuration of the chiral center bearing -NH₂ is (R). However, for 3l with a chiral methyl group at the γ -position, the configuration on the newly generated chiral center bearing -NH₂ is found to be opposite (i.e. S). Similar observation was found in the second part of Fig. 3, the configurations of the chiral centers bearing -NH₂ for 3m-3n and 3r-3t are (R); but for 3o-3q and 3u, they are (S). It looks that the side chain on the neighboring amino acid has a great impact on the selectivity of the biomimetic transamination. The authors are advised to discuss and explain this observation probably by elaborating the proposed mechanism in Fig. 5a.*

Response: Regarding the absolute configurations of the newly generated chiral centers, varying configurations for different transamination products were shown in Fig. 3. The mainly reason is that we utilized chiral pyridoxamine catalyst **1b** with different configurations for different substrates. Previously, the information about the catalyst configuration was included in SI, making the diastereomeric induction confusing. Actually, the catalyst predominantly controlled the stereoselectivity for the transamination, forming the newly generated chiral center with *S* configuration from catalyst (*R*)-**1b** and *R* configuration from (*S*)-**1b**. The chiral groups on the amino acid residues threw ignorable influence on diastereomeric induction probably due to being

far away from the reaction center as well as the flexibility of the skeletons of the α -keto amides. We have added the information about the configuration of chiral pyridoxamine catalyst **1b** used for each substrate in Fig. 3 of the main text. The corresponding discussion has also been added.

(2) *For chiral amino acid esters 2m-u bearing a chiral center, diastereoselectivity is observed using chiral catalyst (S)-1b in the reaction. Considering the possible matched/mismatched transition states in the chiral induction step, would the use of chiral catalyst with opposite chirality (i.e. (R)) affect the diastereoselectivity of the reaction?*

Response: we have investigated the impacts of catalyst and substrates on diastereomeric induction. Several representative α -keto amides were transaminated respectively using (S)-**1b** and (R)-**1b** as the catalyst. The corresponding peptides were formed with *S* configurations from catalyst (R)-**1b** and *R* configurations from (S)-**1b**. The chiral pyridoxamine catalyst dominated the stereoselectivity, while the chiral groups on the amino acid residues threw little influence on the diastereomeric induction probably due to being far away from the reaction center as well as the flexibility of the skeletons of the α -keto amides. No matter which configuration of the catalyst was applied, excellent diastereoselectivities were always obtained, even for α -keto amides (for **3n** and **3aa**) with a nearby bulky chiral amino acid residue. The information and the corresponding discussion have been added to the main text. Figure 3 in the main text has been updated. The corresponding information have also been included in SI.

(3) *Did the authors study substrates bearing chiral amino acid unit together with substrate like **3l** with a chiral methyl group at the γ -position? As both the chiral amino acid and chiral methyl group at the γ -position would affect the diastereoselectivity, it would be interesting to investigate the combined effects of them. Higher, lower or similar diastereoselectivity could be obtained?*

Response: We have examined an α -keto amide **2y** to investigate the combined impact of two nearby chiral centers of α -keto amide on diastereoselectivity. A pair of diastereomers (R,R,S)-**3y** and (R,S,S)-**3y** were obtained both in good yields with high enantiopurities by using (S)-**1b** and (R)-**1b** as the catalyst, respectively. The chiral pyridoxamine catalyst controlled the diastereoselectivity predominantly, while the chiral centers of α -keto amide **2y** threw little influence on the diastereocontrol.

(4) In Fig. 3, the authors have investigated different structures of α -keto amides and different amino acids with diversified side chains, and several peptides have been studied, presenting good tolerance to C-C double bond, NH₂-sensitive bromide, silyl group, OH and NH groups. As presented in Fig. 3, **3q** with a methionine unit can be resulted with good yield and high diastereoselectivity (80% yield, 97:3 dr). Is this reaction compatible with other amino acid side chains, for example, would a cysteine unit with a free thiol group be tolerated?

Response: We have investigated more substrates bearing different functional side chains. The transamination not only well tolerated with functional groups such as NH₂-sensitive bromide (for **3j**), OH group of Tyr (for **3s**) and NH group of Trp (for **3t**), but also displayed good activity and diastereoselectivity with α -keto amides containing a nearby Boc-protected Lys residue (for **3w**), Boc-protected guanidine (for **3x**), or basic NH₂ group of Lys (for **3ab**). The information and the corresponding discussion have been added to the main text. Figure 3 in the main text has been updated. The substrate synthetic procedures, characterization data of these α -keto amides **2** and their transamination products **3** along with NMR spectra have been included in SI.

α -Keto amide **3al** containing a nearby Cys side chain was not obtained due to intramolecular cyclization to form compound methyl (*R*)-5-oxo-6-(2-phenylethylidene)thiomorpholine-3-carboxylate (**3al-cycled**).

Reviewers' Comments:

Reviewer #1:

Remarks to the Author:

The authors have responded adequately to most of my feedback, and it is good to see that they have expanded the substrate scope in the revised manuscript. Still, I question the suitability of this article for this journal, as the substrate scope remains limited in terms of unprotected side chain tolerance, which limits the synthetic utility. Additionally, the novelty is limited in that quaternization of the pyridine N has already been demonstrated by other to enhance transamination chemistry.

Reviewer #2:

Remarks to the Author:

In my opinion the authors have very nicely addressed the points raised by all three previous reviewers and especially the additional experiments that were carried out and which support also the high diastereoselectivities on the peptide-precursors add additional value to this paper. I thus have no further concerns and recommend publication of this article as it stands now

Reviewer #3:

Remarks to the Author:

The authors have well addressed the questions and suggestions in the revised manuscript. The revised manuscript has provided more detailed studies and clearer discussion. Thus, the publication on Nature Communications is recommended.